# Higher red cell distribution width (RDW) is associated with increased all-cause and cardiovascular mortality in patients with breast cancer: A retrospective analysis of NHANES data (1999–2018)

Xuan Liu[1], Hao Wu[2], Saiqian Zhang[3], Zhu Zhu[3], Chao Liu[3], Jinmin Cao[ID][3]*

1 Department of Breast and Thyroid Surgery, The First People's Hospital of Xiangtan City, Hunan Province, China, 2 Department of General Surgery, Hunan Aerospace Hospital, Changsha, Hunan, China, 3 Department of Dermatology, Hunan Aerospace Hospital, Changsha, Hunan, China

* caojinmin_cool@126.com

## Abstract

### Background

The correlation between red cell distribution width (RDW) and mortality in breast cancer participants is not well-defined. This study investigates the association between RDW and both all-cause and cardiovascular mortality in the US population.

### Methods

A retrospective cohort study was performed using data from 15,806 participants in the NHANES dataset. Multivariable Cox regression models were used to analyze demographic, socioeconomic, clinical, and laboratory factors, with adjustments for potential confounders. Restricted cubic spline (RCS) analysis was utilized to investigate the non-linear associations between RDW and mortality outcomes, and Kaplan-Meier (KM) survival curves were created to illustrate RDW's effect on survival rates. Subgroup analyses and time-dependent ROC curves were also utilized to further assess the predictive value of RDW across different time intervals and patient subgroups.

### Results

Elevated RDW significantly correlates with a heightened risk of all-cause mortality (adjusted HR 2.13, 95% CI 1.42–3.20) and cardiovascular mortality (adjusted HR 3.94, 95% CI 1.71–9.09) compared to lower RDW in Model 3. The association remained consistent across subgroups, with no significant interaction effects (p > 0.05). The RCS analysis demonstrated a positive linear relationship between RDW and mortality outcomes. Additionally, Kaplan-Meier analysis indicated that

**Data availability statement:** All data used in this study are publicly available. The primary source is the National Health and Nutrition Examination Survey (NHANES), accessible at https://www.cdc.gov/nchs/nhanes/index.html. The processed analytic dataset (RDW_Breast Cancer data) is provided as Supporting Information File S1 and is also available on Figshare at https://figshare.com/articles/dataset/29207858 with DOI 10.6084/m9.figshare.29207858.

**Funding:** No specific funding was received for this work.

**Competing interests:** The authors have declared that no competing interests exist.

**Abbreviations:** NHANES, National Health and Nutrition Examination Survey; RDW, red cell distribution width; ALB, albumin; BMI, Body mass index; PIR, Poverty income ratio; TC, total cholesterol; HDL: high-density lipoprotein cholesterol; HbA1c, Glycosylated hemoglobin; HGB, hemoglobin; MCV: mean corpuscular volume; eGFR: estimated glomerular filtration rate; NCHS, National Center for Health Statistics; CDC, Centers for Disease Control and Prevention; HR, hazard ratio; CI, confidence interval.

individuals with elevated RDW levels exhibited significantly lower survival rates. Time-dependent ROC and AUC analyses demonstrated that RDW was a more robust predictor of short-term mortality, as evidenced by higher AUC values in the initial years following diagnosis.

## Conclusions

Red cell distribution width (RDW) serves as an independent predictor of both all-cause and cardiovascular mortality in breast cancer patients, showing strong predictive power for outcomes in both the short and long term.

## Introduction

Breast cancer (BC) represents about 32% of all newly diagnosed cancers in women, making it the most prevalent cancer type among this population. Since the mid-2000s, its incidence rate has been increasing by 0.6% annually. Breast cancer is the leading cause of cancer-related deaths in women between the ages of 20 and 49 [1]. In recent decades, advancements in screening methods and treatment options for breast cancer have led to improved survival rates specific to the disease [2]. Complications such as metastasis to vital organs, lymphedema, and treatment-induced cardiovascular diseases contribute to the overall burden and increase mortality rates [3,4]. Recent studies have found that cardiovascular mortality among breast cancer patients is also gradually increasing [5,6]. Hence, investigating alternative markers, such as routine blood tests, is essential, as they offer rapid and simple insights that can help healthcare providers assess the prognosis of breast cancer patients.

Red cell distribution width (RDW) is a blood parameter that reflects the variation in the size of red blood cells, a condition referred to as anisocytosis. Originally applied in anemia diagnostics, RDW has become a notable prognostic marker in numerous diseases because of its links to inflammation and adverse clinical outcomes [7,8]. Increased RDW are linked to a range of conditions, including heart-related diseases (such as heart failure and coronary artery disease) [9–12], sepsis [13,14] and acute respiratory distress syndrome [15,16]. Recent studies indicate that increased RDW is linked to prognosis in various tumors, including mortality in colorectal [17], laryngeal [18], and esophageal cancers [19]. These results suggest that red cell distribution width may act as a new biomarker for breast cancer activity [20]. In a study involving 203 breast cancer patients under 40, pretreatment RDW showed a potential association with both disease-free survival (DFS) and overall survival (OS).

In a study conducted by Takeuchi H et al. involving 299 breast cancer patients, a connection was found between the red cell distribution width to platelet ratio and the prognosis of breast cancer [21]. Similarly, a cohort study involving 825 patients revealed that elevated pretreatment RDW levels correlated with increased lymphatic metastasis and reduced disease-free survival (DFS) and overall survival (OS) [22]. Current research shows no consensus on RDW's impact on breast cancer patients.

This study aims to examine the link between RDW and breast cancer mortality using a substantial outpatient adult sample from the NHANES database.

## Materials and methods

### Study design and participants

This study employed data from the US National Health and Nutrition Examination Survey (NHANES), an extensive, continuous survey by the US Centers for Disease Control and Prevention (CDC). NHANES employs a stratified, multi-stage probability sampling method to collect health and nutritional information representative of the non-institutionalized US population. The dataset, covering 1999–2018, includes diverse health metrics such as demographic information, physical examinations, and laboratory tests. NHANES data are publicly accessible, with all participants giving informed consent under protocols sanctioned by the NCHS Ethics Review Committee. Access to the NHANES repository is open via the CDC website (https://www.cdc.gov/nchs/nhanes/index.html).

The study utilized data sourced from the NHANES database. We obtained publicly accessible data spanning 20 years (1999–2018) from the NHANES website. Across ten survey cycles, 101,316 participants completed the surveys. Following the exclusion of 49,893 males, 22,815 individuals under 20, 1,541 pregnant individuals, and 20,169 participants with incomplete data on RDW, breast cancer, mortality, or other covariates, the final analysis included a cohort of 158,06 participants. The participant selection process is detailed in S1 Fig.

### Ascertainment of breast cancer and all-cause mortality and cardiovascular mortality

The diagnosis of breast cancer (BC) in NHANES was determined through self-reported data collected from the medical condition surveys. Participants were queried about whether a healthcare provider had ever diagnosed them with cancer or any malignancy. Those who responded "yes" were further asked to specify the type of cancer. Individuals diagnosed solely with breast cancer as their primary and only tumor were classified as BC cases. Participants who either answered "no," reported other types of cancer, or had a history of breast cancer along with other cancers were categorized as non-BC individuals [23]. Mortality data for the study population were sourced through linkage between NHANES and the National Death Index (NDI), available at https://www.cdc.gov/nchs/data-linkage/mortality.htm. Participants were classified as either alive or deceased based on NDI records. Each participant's follow-up period was monitored from their NHANES examination date until the earlier of their date of death or December 31, 2019. Death causes were classified according to the 10th revision of the International Statistical Classification of Diseases (ICD-10). Cardiovascular-related deaths were classified using ICD-10 codes I00-I09, I11, I13, and I20-I51, as defined by the National Center for Health Statistics (NCHS) [24].

### Ascertainment of RDW

In our study, RDW was measured only at baseline during the initial examination visit, and no repeated measurements were conducted over time. RDW determination was performed based on the full blood count data, utilizing a Beckman Coulter MAXM automated hematology analyzer to measure the values. The procedure strictly followed the guidelines detailed in the NHANES laboratory protocols designed for medical technologists. The concentration of albumin, a critical marker for assessing nutritional health and a significant contributor to the maintenance of colloidal osmotic pressure, was determined employing the DcX800 method. This approach utilizes a dual-wavelength digital endpoint technique for accurate measurement.

### Covariates

Covariates, encompassing markers for baseline characteristics, physical examinations, medications and therapies, comorbidities, and laboratory values, were derived from previous studies. The baseline variables comprised age,

sex, race, education, marital status, PIR, BMI, hypertension, coronary heart disease, angina, heart failure, diabetes, family history of diabetes, hyperlipidemia, and hormone replacement therapy. Laboratory assessments included HbA1c, total cholesterol, high-density lipoprotein, albumin, hemoglobin, mean corpuscular volume (MCV), serum creatinine, and estimated glomerular filtration rate (eGFR). A diagnosis of diabetes is confirmed either through a patient's report of a physician's diagnosis or by fulfilling at least one of the subsequent criteria: a glycated hemoglobin (HbA1c) concentration of 6.5% or higher, a fasting blood glucose level of 126 mg/dL or more, a glucose level of 200 mg/dL or higher after a 2-hour oral glucose tolerance test (OGTT), or a random blood glucose level of 200 mg/dL or more in the presence of characteristic symptoms [25]. Hypertension is identified when a patient's systolic blood pressure reaches 140 mmHg or higher, diastolic pressure is 90 mmHg or higher, they are on antihypertensive medication, or when confirmed by a physician's report [26]. Coronary heart disease was identified through a positive response to the question: "Has your physician or other healthcare professional informed you that you have coronary heart disease? "Hyperlipidemia was diagnosed if participants reported a physician's diagnosis, were taking lipid-lowering medications, or had a total cholesterol level ≥ 240 mmol/L. Participants were categorized as hormone replacement therapy (HRT) users if they affirmed using female hormones (e.g., estrogen or progesterone) for menopause-related symptoms, mental health issues, post-hysterectomy or oophorectomy, osteoporosis or cardiovascular disease prevention, menstrual regulation, or other reasons [27]. The laboratory assessments included red blood cell width, albumin, hemoglobin, MCV, glycosylated hemoglobin (HbA1c), total cholesterol, and eGFR. The eGFR was estimated using the Chronic Kidney Disease Epidemiology Collaboration formula [28] and categorized as either ≥ 60 or < 60.

## Statistical analysis

The Free Statistics Analysis Platform version 1.9 is a user-friendly software application designed for conducting standard statistical analyses and creating data visualizations. This platform is powered by R, which serves as the underlying statistical engine, with the user interface developed in Python. For continuous data that follows a normal distribution, the results are presented as the mean along with the standard deviation. In cases where the data is not normally distributed, the median and interquartile range are provided. For categorical or binary variables, the data is expressed in terms of relative frequencies. For the comparison of continuous data, we employed the Student's t-test, whereas the chi-square ($\chi^2$) test was used to analyze categorical variables. Additionally, Cox proportional hazards regression models were applied to evaluate the correlation between red cell distribution width (RDW) and the incidence of breast cancer, taking into account various clinical features.

   Three distinct analytical models were constructed to account for various demographic and socioeconomic factors. The model 1 incorporated adjustments for factors such as age, race, marital status, and the poverty income ratio (PIR). Model 2 incorporated these variables and further adjusted for BMI, hypertension, coronary heart disease, angina, heart failure, diabetes, family history of diabetes, hormone replacement therapy, and hyperlipidemia; Model 3 encompassed all the variables present in Model 2 and further refined the analysis by incorporating additional factors such as glycated hemoglobin (HbA1c), overall cholesterol levels, high-density lipoprotein cholesterol (HDL-C), serum albumin, hemoglobin concentration, mean corpuscular volume (MCV), serum creatinine levels, and eGFR.

   The relationship between RDW and the risk of all-cause or cardiovascular mortality among breast cancer patients was explored using Restricted cubic spline (RCS) analysis. The Kaplan-Meier estimator was utilized to calculate survival probabilities, with the log-rank test employed for comparing survival curves. A time-dependent receiver operating characteristic (ROC) analysis was performed to assess the predictive value of red cell distribution width (RDW) concerning survival. Data with missing covariate values were excluded from our analysis. The threshold for statistical significance was set at a p-value of less than 0.05.

## Results

### Participants and baseline characteristics

The study included 15,806 participants, categorized into low RDW (<12.68%) and high RDW (≥12.68%) groups, with 6,513 and 9,293 participants, respectively. Individuals with high RDW exhibited a significantly greater average age of 51.3 ± 17.7 years compared to those with low RDW, whose mean age was 46.2 ± 17.3 years (p < 0.001). Individuals exhibiting elevated red cell distribution width (RDW) tended to have less education, a higher body mass index (BMI), and a higher prevalence of comorbidities such as hypertension, diabetes, and cardiovascular conditions. The group with elevated red cell distribution width (RDW) experienced a notably higher rate of all-cause mortality at 14.8% compared to the group with low RDW, which had a rate of 10.8% (p < 0.001). Additionally, the elevated RDW group had a reduced follow-up period, averaging 109.9 ± 58.6 months, in contrast to the low RDW group with a mean follow-up of 151.1 ± 54.6 months (p < 0.001). The study included 15,806 participants, with 476 diagnosed with breast cancer. Participants with breast cancer were older and exhibited a higher prevalence of comorbidities, including elevated RDW, hypertension, diabetes, and hyperlipidemia, compared to those without the disease (Table 1, S1 Table).

Continuous variables are presented as the mean and 95% confidence interval, category variables are described as the percentage and 95% confidence interval.

### Associations between RDW and all-cause mortality in participants with breast cancer

The study investigates the relationship between red cell distribution width (RDW) and mortality outcomes in breast cancer participants, including all-cause and cardiovascular deaths across various regression models. Elevated red cell distribution width (RDW) was associated with an increased likelihood of all-cause mortality, with a hazard ratio (HR) of 1.39 (95% CI 1.25–1.54, p < 0.001) in the unadjusted model, rising to 1.56 (95% CI 1.35–1.8, p < 0.001) after full adjustment in Model 3.Elevated RDW are significantly linked to a higher risk of all-cause mortality, with an adjusted hazard ratio of 2.13 (95% CI 1.42–3.2, p < 0.001) (Table 2).Fig 1 illustrates the results of Cox proportional hazards models, showing a linear relationship between RDW and the risk of all-cause mortality among breast cancer patients, although this was not statistically significant (p = 0.442). The analysis using restricted cubic splines demonstrates a direct correlation, indicating that a rise in RDW is linked with an escalation in hazard ratios for all-cause mortality. The research revealed a direct relationship, suggesting that higher RDW are associated with a greater risk of all-cause mortality among individuals with breast cancer (Fig 1A).

Fig 2 displays Kaplan-Meier survival curves demonstrating the effect of RDW on all-cause mortality, comparing RDW levels below 12.68% with those at or above 12.68%. The data indicates that patients with RDW levels of 12.68% or higher exhibit a notably reduced survival rate compared to those with levels below 12.68%. The results indicate that higher RDW correlates with poorer prognosis and may be an effective predictor of mortality risk in this patient group (Fig 2A).The subgroup analysis evaluated the
association between RDW and the risk of all-cause mortality across various clinical subgroups within the breast cancer patient population. The results consistently showed that elevated RDW was significantly linked to higher all-cause mortality, particularly in older patients (≥65 years).The association's robustness was validated across various models, with no significant interaction effects (p > 0.05), demonstrating consistent results in all stratified analyses (Fig 3).

### Associations between RDW and cardiovascular mortality in patients with breast cancer

Fig 1B illustrates the Restricted Cubic Splines (RCS) analysis, demonstrating a linear relationship between RDW and cardiovascular mortality (p = 0.245).The graph indicates a significant correlation between elevated RDW values and increased cardiovascular mortality. The hazard ratio (HR) increased from 1.48 (95% CI: 1.21–1.8, p < 0.001) in the unadjusted model to 1.79 (95% CI: 1.44–2.23, p < 0.001) in the fully adjusted model. In the fully adjusted model, high RDW significantly

**Table 1. Baseline characteristics of critical patients with low and high RDW.**

| Characteristic | Total | Low RDW (<12.68%) | High RDW (≥12.68%) | p |
|---|---|---|---|---|
| Sample size, N | 15806 | 6513 | 9293 | |
| Age(years) | 49.2±17.7 | 46.2±17.3 | 51.3±17.7 | < 0.001 |
| Race, n (%) | | | | < 0.001 |
| Non-Hispanic White | 2831 (17.9) | 1244 (19.1) | 1587 (17.1) | |
| Non-Hispanic Black | 1357 (8.6) | 487 (7.5) | 870 (9.4) | |
| Mexican American | 7141 (45.2) | 3508 (53.9) | 3633 (39.1) | |
| Other | 4477 (28.3) | 1274 (19.6) | 3203 (34.5) | |
| Education, n (%) | | | | < 0.001 |
| below high school | 3987 (25.2) | 1459 (22.4) | 2528 (27.2) | |
| high school | 3527 (22.3) | 1419 (21.8) | 2108 (22.7) | |
| above high school | 8292 (52.5) | 3635 (55.8) | 4657 (50.1) | |
| Marital, n (%) | | | | < 0.001 |
| Married | 8648 (54.7) | 3857 (59.2) | 4791 (51.6) | |
| Living alone | 4501 (28.5) | 1523 (23.4) | 2978 (32) | |
| Never married | 2657 (16.8) | 1133 (17.4) | 1524 (16.4) | |
| PIR, n (%) | | | | < 0.001 |
| Low | 5039 (31.9) | 1756 (27) | 3283 (35.3) | |
| Medium | 5966 (37.7) | 2395 (36.8) | 3571 (38.4) | |
| High | 4801 (30.4) | 2362 (36.3) | 2439 (26.2) | |
| BMI (kg/m$^2$) | 29.4±7.4 | 27.6±6.1 | 30.6±8.0 | < 0.001 |
| Hypertensive, n (%) | | | | < 0.001 |
| No | 9225 (58.4) | 4304 (66.1) | 4921 (53) | |
| Yes | 6581 (41.6) | 2209 (33.9) | 4372 (47) | |
| Heart failure, n (%) | | | | < 0.001 |
| No | 15427 (97.6) | 6451 (99) | 8976 (96.6) | |
| Yes | 379 (2.4) | 62 (1) | 317 (3.4) | |
| Coronary heart disease, n(%) | | | | < 0.001 |
| No | 15426 (97.6) | 6416 (98.5) | 9010 (97) | |
| Yes | 380 (2.4) | 97 (1.5) | 283 (3) | |
| Angina, n (%) | | | | < 0.001 |
| No | 15433 (97.6) | 6396 (98.2) | 9037 (97.2) | |
| Yes | 373 (2.4) | 117 (1.8) | 256 (2.8) | |
| Diabetes, n (%) | | | | < 0.001 |
| No | 13451 (85.1) | 5828 (89.5) | 7623 (82) | |
| Yes | 2355 (14.9) | 685 (10.5) | 1670 (18) | |
| Diabetes family history, n (%) | | | | < 0.001 |
| No | 8356 (52.9) | 3567 (54.8) | 4789 (51.5) | |
| Yes | 7450 (47.1) | 2946 (45.2) | 4504 (48.5) | |
| Hyperlipidemia, n(%) | | | | < 0.001 |
| No | 9738 (61.6) | 4227 (64.9) | 5511 (59.3) | |
| Yes | 6068 (38.4) | 2286 (35.1) | 3782 (40.7) | |
| Hormone replacement therapy, n (%) | | | | < 0.001 |
| No | 14261 (90.2) | 5631 (86.5) | 8630 (92.9) | |
| Yes | 1545 (9.8) | 882 (13.5) | 663 (7.1) | |

*(Continued)*

**Table 1.** (Continued)

| Characteristic | Total | Low RDW (<12.68%) | High RDW (≥12.68%) | p |
|---|---|---|---|---|
| MCV, Mean±SD | 88.8±6.1 | 91.0±4.1 | 87.3±6.8 | < 0.001 |
| Total cholesterol(mg/dL) | 198.4±41.0 | 198.9±39.8 | 198.0±41.7 | 0.195 |
| High density lipoprotein (mg/dL) | 57.3±16.1 | 57.6±15.6 | 57.0±16.5 | 0.029 |
| HbA1c (%) | 5.7±1.1 | 5.5±1.0 | 5.8±1.1 | < 0.001 |
| Hemoglobin (mg/dL) | 13.4±1.3 | 13.7±1.0 | 13.1±1.3 | < 0.001 |
| Albumin, (mg/dL) | 4.2±0.3 | 4.2±0.3 | 4.1±0.3 | < 0.001 |
| Serum creatinine (mg/dL) | 0.7 (0.6, 0.8) | 0.7 (0.6, 0.8) | 0.8 (0.7, 0.9) | < 0.001 |
| eGFR, Mean±SD | 95.6±23.1 | 99.5±20.9 | 92.8±24.2 | < 0.001 |
| Breast cancer, n (%) | | | | < 0.001 |
| No | 15330 (97.0) | 6376 (97.9) | 8954 (96.4) | |
| Yes | 476 (3.0) | 137 (2.1) | 339 (3.6) | |
| All-cause mortality, n (%) | | | | < 0.001 |
| No | 13722 (86.8) | 5809 (89.2) | 7913 (85.2) | |
| Yes | 2084 (13.2) | 704 (10.8) | 1380 (14.8) | |
| CVD mortality, n (%) | | | | < 0.001 |
| No | 15144 (95.8) | 6305 (96.8) | 8839 (95.1) | |
| Yes | 662 (4.2) | 208 (3.2) | 454 (4.9) | |
| Follow-up time(months) | 126.9±60.5 | 151.1±54.6 | 109.9±58.6 | < 0.001 |

PIR: Poverty income ratio; BMI: body mass index; CVD: cardiovascular disease; FHD: Family history of diabetes; TC: Total cholesterol; HDL: high-density lipoprotein cholesterol; SCR: serum creatinine; HbA1c: Glycosylated hemoglobin; Hb: hemoglobin; eGFR: estimated glomerular filtration rate; MCV, mean corpuscular volume; RDW, red cell distribution width;

elevated the risk of cardiovascular mortality, with an HR of 3.94 (95% CI 1.71–9.09, p = 0.001) (Table 2). Fig 2B's Kaplan-Meier survival curves highlight the differences in cardiovascular mortality between patients with low and high RDW levels. Patients with high RDW (≥12.68%) had significantly lower cardiovascular survival rates throughout the follow-up period. After around 50 months of follow-up, the high RDW group consistently exhibits poorer survival outcomes than the low RDW group, highlighting a notable separation between the curves.

**ROC analysis of the predictive value of the RDW for all-cause and cardiovascular mortality in patients with breast cancer**

The analysis utilized time-dependent ROC curves to assess the predictive accuracy of RDW at various time points. The 1-year Area Under the Curve (AUC) for predicting all-cause mortality was 0.85. The AUC values were 0.71 at 3 years, 0.63 at 5 years, and 0.71 at 10 years. Similarly, the predictive performance of RDW for cardiovascular mortality showed AUC values of 0.85 at 1 year, 0.69 at 3 years, 0.66 at 5 years, and 0.69 at 10 years, each with their respective 95% CIs. RDW showed strong predictive capability for all-cause and cardiovascular mortality, especially in the short term. RDW is a more effective predictor of short-term outcomes compared to long-term outcomes in this patient cohort (Fig 4).

## Discussion

The research highlights a substantial correlation between elevated RDW values and heightened risks of overall and cardiovascular mortality among breast cancer patients. In a cohort of 939 women with breast cancer, elevated RDW demonstrated strong independent prognostic value for all-cause mortality within the first 6 months, 1 year, and 3 years following

**Table 2. Association between RDW and RDW group and breast cancer in multiple regression model.**

| Variable | HR (95%CI), P value | | | |
|---|---|---|---|---|
| | Crude | Model 1 | Model 2 | Model 3 |
| **All-cause mortality** | | | | |
| RDW | 1.39 (1.25~1.54)<0.001 | 1.34 (1.2~1.5)<0.001 | 1.37 (1.22~1.54)<0.001 | 1.56 (1.35~1.8)<0.001 |
| RDW group | | | | |
| Low RDW (<12.68%) | Reference | Reference | Reference | Reference |
| High RDW (≥12.68%) | 2.39 (1.66~3.45) <0.001 | 1.9 (1.31~2.76) 0.002 | 2.14 (1.44~3.18) <0.001 | 2.13 (1.42~3.2) <0.001 |
| **Cardiovascular mortality** | | | | |
| RDW | 1.48 (1.21~1.8) <0.001 | 1.43 (1.13~1.77) 0.002 | 1.44 (1.14~1.83)0.003 | 1.79 (1.44~2.23) <0.001 |
| RDW group | | | | |
| Low RDW (<12.68%) | Reference | Reference | Reference | Reference |
| High RDW (≥12.68%) | 3.59 (1.58~8.16)0.002 | 2.8 (1.2~6.53)0.013 | 3.87 (1.54~9.72)0.004 | 3.94 (1.71~9.09)0.001 |

Notes: Crude model: Unadjusted model;

Model 1: adjusted for sociodemographic variables (Age, Race, Martial status, PIR);

Model 2: Model 1 and BMI, Hypertensive, Cardiovascular disease, angina, heart failure, diabetes, family history of diabetes, Hyperlipidemia,

Model 3: adjusted for Model2, HbA1c, Total cholesterol, High density lipoprotein, Albumin, Hemoglobin, MCV, Serum creatinine, eGFR.

Abbreviations: RDW, red cell distribution width; BMI, Body mass index; PIR, Poverty income ratio; TC, total cholesterol; HbA1c, Glycosylated hemoglobin; Hb, hemoglobin; MCV: mean corpuscular volume; eGFR: estimated glomerular filtration rate;HR, hazard ratio.

identification [29]. Furthermore, among a group of 4,884 individuals with breast cancer, elevated RDW was notably linked to larger tumor dimensions, later disease stages, and a higher incidence of lymph node involvement. Increased RDW has been linked to poorer overall and disease-free survival rates [22]. A retrospective study of 299 breast cancer patients in Japan, however, found no correlation between RDW and disease-free survival (DFS). These discrepancies may arise from variations in study populations, sample sizes, and cancer stages [21].

These mechanisms, both individually and in combination, may affect breast cancer progression and mortality: 1) Elevated RDW is frequently associated with chronic systemic inflammation [8]. Increased concentrations of inflammatory cytokines such as interleukin-6 (IL-6) and tumor necrosis factor-α (TNF-α) in breast cancer patients promote tumor progression, angiogenesis, and invasion. Breast cancer progression is significantly influenced by inflammation and immune dysregulation [30]. Cytokines disrupt iron metabolism and inhibit erythropoietin production, resulting in greater variation in red blood cell size and higher RDW levels [31]. Furthermore, breast cancer patients frequently exhibit immune dysregulation, as indicated by elevated neutrophil levels and aberrant immune responses, which compromise the body's capacity to combat tumor cells [32]. The immunosuppression and chronic inflammation that accompany this process result in oxidative stress, which further damages red blood cells and increases RDW. 2)Oxidative stress plays a key role in the initiation and progression of breast cancer. Increased reactive oxygen species (ROS) levels cause damage to DNA, proteins, and lipids, facilitating cancer cell growth and spread [33]. This oxidative stress also impairs red blood cell (RBC) membrane integrity, shortening their lifespan and leading to greater RBC variability and higher RDW [34]. Patients with breast cancer frequently exhibit heightened oxidative stress, which is linked to systemic inflammation and nutritional deficiencies [35]. 3)Nutrient deficiencies are prevalent among individuals diagnosed with breast cancer, particularly those deficient in iron, vitamin B12, and folic acid [36]. These nutrients are essential for the synthesis of hemoglobin and the production of DNA. Deficiencies in these nutrients can lead to the formation of larger, immature erythrocytes, which can result in increased erythrocyte size variability and elevated RDW [37]. Chemotherapy and other cancer treatments typically interfere with

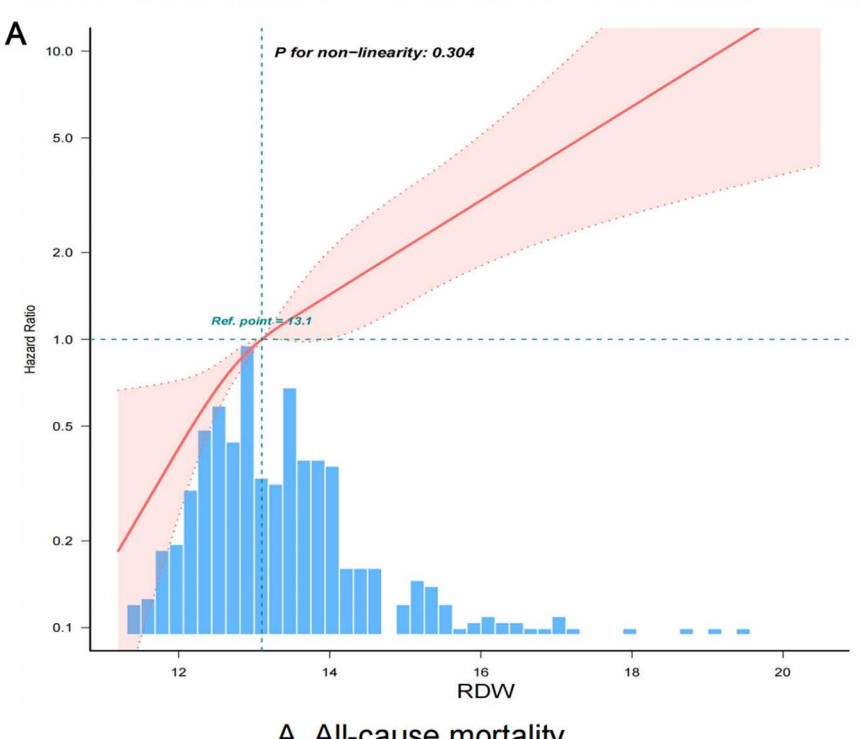

A  All-cause mortality

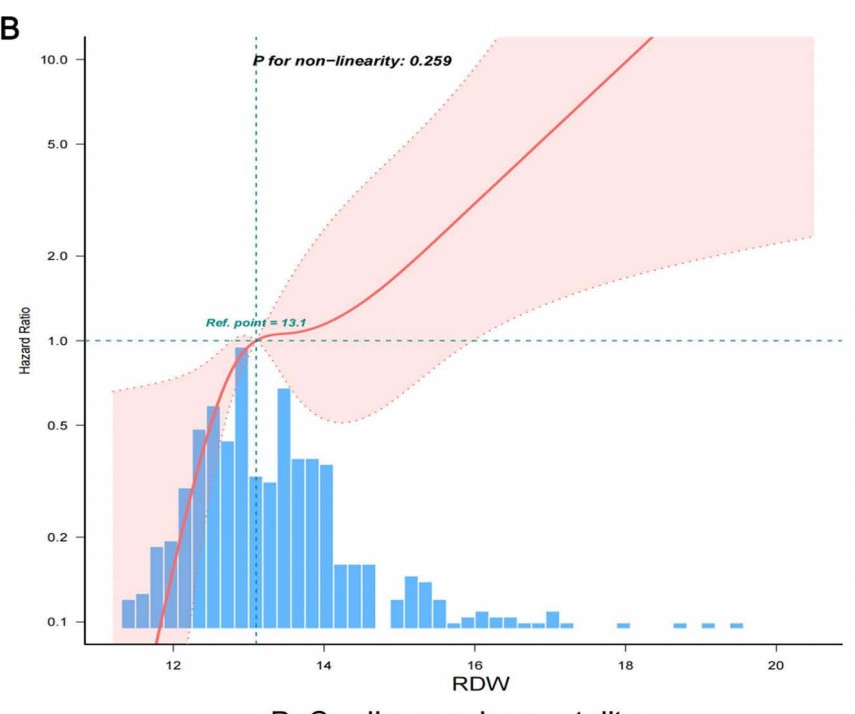

B  Cardiovascular mortality

**Fig 1. A linear relationship between RDW with all-cause (A) and cardiovascular mortality (B) among breast cancer visualized by restricted cubic spline.** The association of RDW with all-cause (A) and cardiovascular mortality (B) among breast cancer visualized by restricted cubic spline. Hazard ratios were adjusted age, race, marital status, PIR, BMI, hypertension, diabetes, hyperlipidemia, heart failure, cardiovascular disease, angina, hormone replacement therapy, diabetes family history. MCV, HbA1c, Total cholesterol, High density lipoprotein, Serum creatinine, eGFR. Abbreviation: RDW, red blood cell distribution width;.

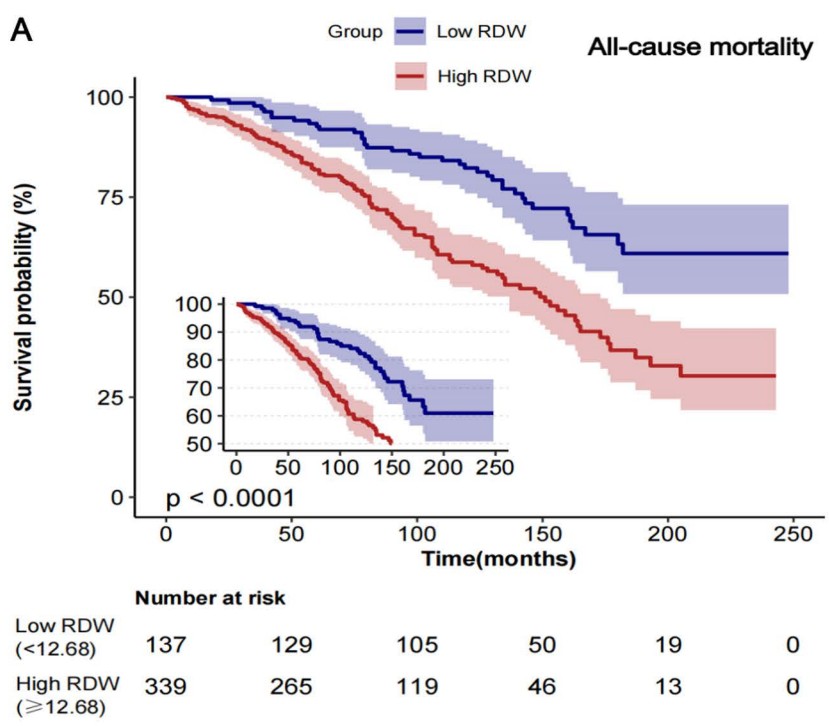

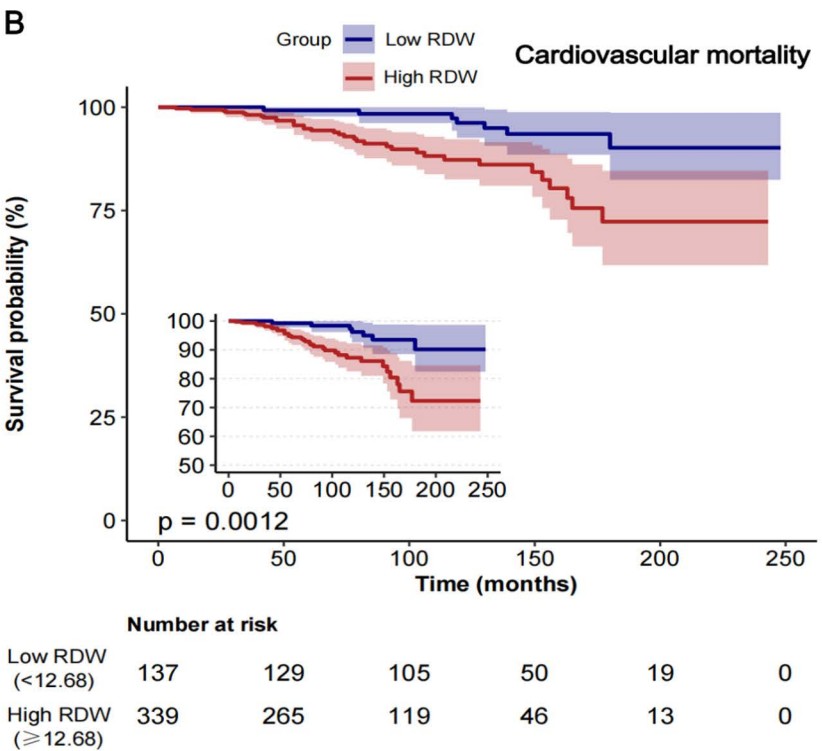

**Fig 2. Kaplan–Meier curves of the survival rate with high (<12.68%) and low (≥12.68%) RDW values.** A. All-cause mortality; B. Cardiovascular mortality. Abbreviation: RDW, red blood cell distribution width.

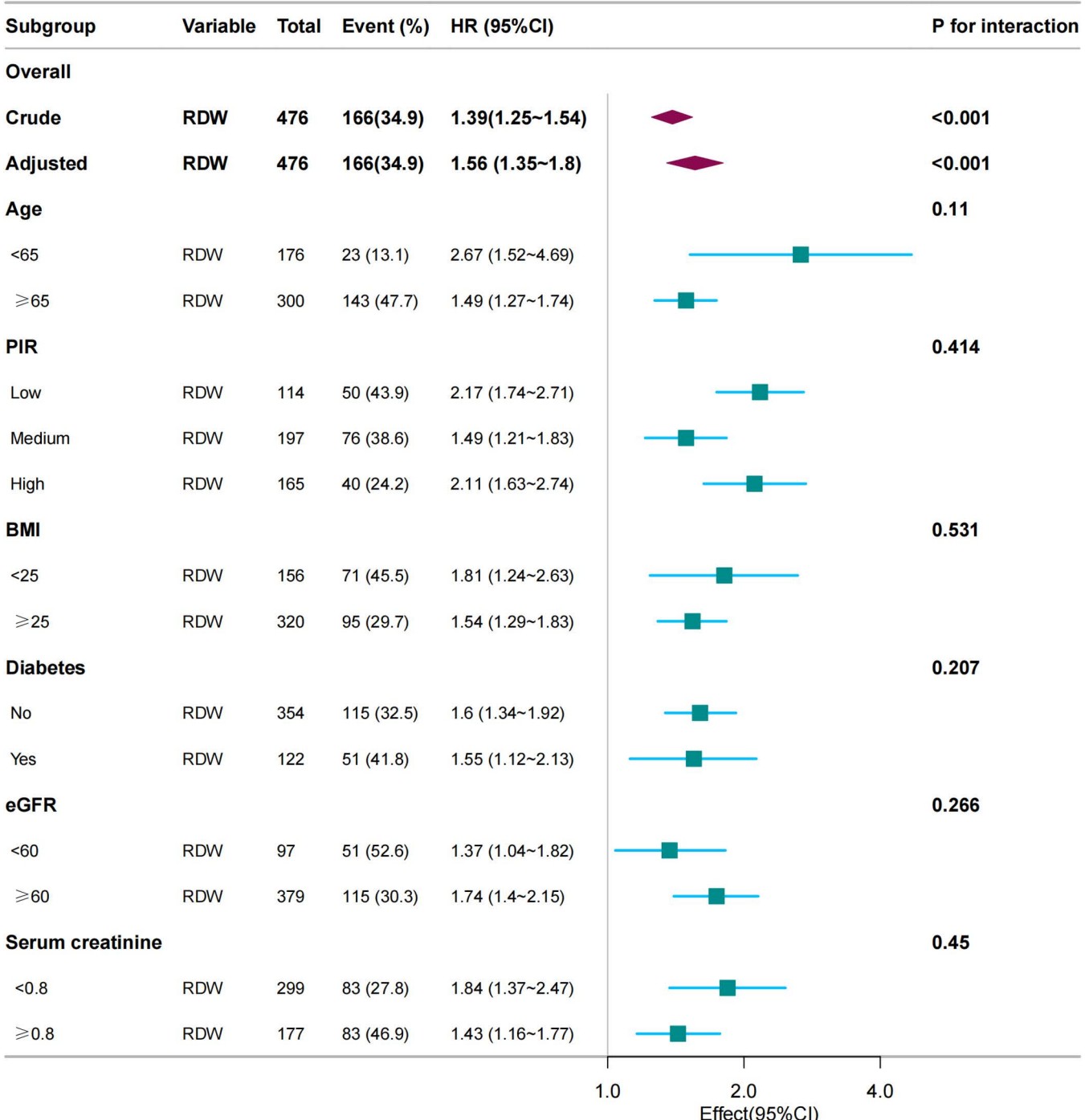

| Subgroup | Variable | Total | Event (%) | HR (95%CI) | | P for interaction |
|---|---|---|---|---|---|---|
| **Overall** | | | | | | |
| **Crude** | **RDW** | **476** | **166(34.9)** | **1.39(1.25~1.54)** | | **<0.001** |
| **Adjusted** | **RDW** | **476** | **166(34.9)** | **1.56 (1.35~1.8)** | | **<0.001** |
| **Age** | | | | | | **0.11** |
| <65 | RDW | 176 | 23 (13.1) | 2.67 (1.52~4.69) | | |
| ≥65 | RDW | 300 | 143 (47.7) | 1.49 (1.27~1.74) | | |
| **PIR** | | | | | | **0.414** |
| Low | RDW | 114 | 50 (43.9) | 2.17 (1.74~2.71) | | |
| Medium | RDW | 197 | 76 (38.6) | 1.49 (1.21~1.83) | | |
| High | RDW | 165 | 40 (24.2) | 2.11 (1.63~2.74) | | |
| **BMI** | | | | | | **0.531** |
| <25 | RDW | 156 | 71 (45.5) | 1.81 (1.24~2.63) | | |
| ≥25 | RDW | 320 | 95 (29.7) | 1.54 (1.29~1.83) | | |
| **Diabetes** | | | | | | **0.207** |
| No | RDW | 354 | 115 (32.5) | 1.6 (1.34~1.92) | | |
| Yes | RDW | 122 | 51 (41.8) | 1.55 (1.12~2.13) | | |
| **eGFR** | | | | | | **0.266** |
| <60 | RDW | 97 | 51 (52.6) | 1.37 (1.04~1.82) | | |
| ≥60 | RDW | 379 | 115 (30.3) | 1.74 (1.4~2.15) | | |
| **Serum creatinine** | | | | | | **0.45** |
| <0.8 | RDW | 299 | 83 (27.8) | 1.84 (1.37~2.47) | | |
| ≥0.8 | RDW | 177 | 83 (46.9) | 1.43 (1.16~1.77) | | |

Effect(95%CI)

**Fig 3. Forest plot depicting the association between the RDW and the risk of all-cause mortality.** Abbreviation: PIR: Poverty income ratio; BMI: body mass index; eGFR: estimated glomerular filtration rate; RDW, red blood cell distribution width.

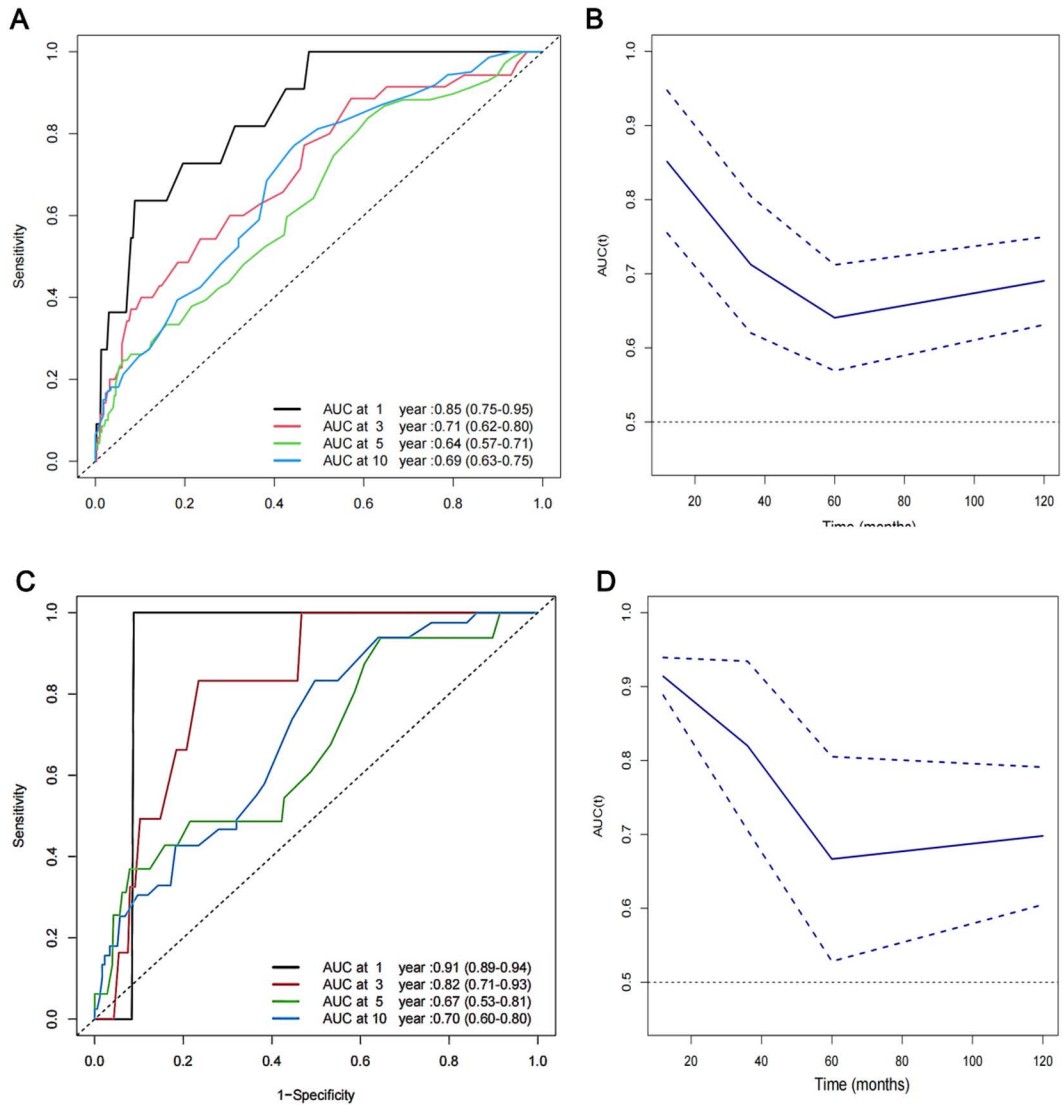

**Fig 4. Time-dependent ROC curves and time-dependent AUC values.** Time-dependent ROC curves and time-dependent AUC values (with 95% confidence band) of the RDW for predicting all-cause mortality (A, B), cardiovascular mortality (C, D).

nutrient absorption and appetite, thus exacerbating these nutrient deficiencies and further exacerbating anemia in breast cancer patients. Anemia results in tissue hypoxia, which subsequently drives tumor progression and treatment resistance, thereby associating elevated RDW with a worse prognosis in breast cancer [38]. 4) Breast cancer survivors are at an increased risk of developing cardiovascular disease (CVD) [5,6]. Additionally, therapies such as anthracyclines and radiotherapy can have harmful effects on the heart, potentially leading to issues like heart failure, cardiomyopathy, and coronary artery disease [39]. Radiation therapy, especially for the left breast, can cause damage to the heart and blood vessels, further increasing cardiovascular risk in breast cancer survivors [40]. Radiation therapy, especially to the left breast, can damage the heart and blood vessels, increasing the cardiovascular risk for breast cancer survivors. As a result, cardiovascular disease has become a major cause of mortality, particularly in older patients [41]. Elevated RDW is linked to increased inflammation, oxidative stress, and impaired erythropoiesis, contributing to cardiovascular

complications and mortality [10,12,42]. These mechanisms contribute to both tumor progression and metastasis, while also heightening the risk of cardiovascular complications in breast cancer patients.

This study has several notable strengths, including a large sample size of 15,806 participants from the NHANES dataset, which enhances the statistical power and generalizability of the findings to the broader US population. The retrospective cohort design enables analysis of long-term outcomes related to RDW in breast cancer patients. Multivariable Cox regression models adjust for multiple factors to reduce confounding, while restricted cubic spline (RCS) analysis identifies potential non-linear relationships between RDW and mortality. Kaplan-Meier and time-dependent ROC curves evaluate RDW as a mortality predictor, especially in the short term. Consistent results across subgroups further support RDW's reliability as a predictor. In today's era of big data, the simplicity of RDW allows for rapid risk stratification prior to the deployment of expensive/invasive tests. RDW can be effectively integrated with other extensive datasets to enhance the accuracy and reliability of predictive models.

The limitations of this study include the inability to establish a definitive causal link between elevated RDW and mortality in breast cancer patients. Despite demonstrating a significant association between RDW and mortality, our observational study's findings are limited by potential residual confounding and the inability to establish causality due to unmeasured treatment details and temporal changes. Secondly, despite adjusting for various confounding factors, residual confounding effects may persist, including those related to cancer treatment modalities, comorbidities and RBC transfusion history-particularly relevant given established transfusion-induced RDW fluctuations [43]. Thirdly, measuring RDW at a single time point may not capture temporal fluctuations that could influence mortality risk. Furthermore, the study population was derived from NHANES data, which may restrict the generalizability of the findings to other populations with disparate characteristics.

## Conclusion

RDW is an important independent indicator of both all-cause and cardiovascular mortality in breast cancer patients, emphasizing its potential as a simple and cost-effective biomarker for clinical risk assessment.

### AI usage statement

The manuscript has been polished with the aid of AI-driven language enhancement tools like ChatGPT(OpenAI GPT-4 version), which has elevated the text's clarity and fluency. Nonetheless, the core elements such as conceptual development, data analysis, and the interpretation of findings remain the intellectual property and original work of the authors.

### Supporting information

**S1 Fig. The flow chart of the study.**
(DOCX)

**S1 Table. General characteristics of the study population according to breast cancer.**
(DOCX)

**S1 Data. RDW_Breast Cancer data.** Original data for this study. https://figshare.com/account/articles/29207858?-file=55028801. The DOI for this dataset is:10.6084/m9.figshare.29207858.
(CSV)

### Acknowledgments

We would like to express our appreciation to Jie Liu from the Department of Vascular and Endovascular Surgery and Huanxian Liu from the Department of Neurology at the First Medical Center, Chinese PLA General Hospital, Beijing, China, for their statistical assistance and insightful feedback on the manuscript.

## Author contributions

**Conceptualization:** Xuan Liu, Hao Wu, Zhu Zhu, Chao Liu.

**Data curation:** Jinmin Cao, Xuan Liu, Hao Wu.

**Formal analysis:** Jinmin Cao, Xuan Liu, Hao Wu, Zhu Zhu.

**Investigation:** Jinmin Cao, Xuan Liu.

**Methodology:** Jinmin Cao, Zhu Zhu, Chao Liu.

**Project administration:** Jinmin Cao.

**Resources:** Jinmin Cao.

**Software:** Jinmin Cao.

**Supervision:** Jinmin Cao.

**Visualization:** Jinmin Cao, Xuan Liu, Hao Wu, Zhu Zhu.

**Writing – original draft:** Jinmin Cao, Xuan Liu, Hao Wu, Saiqian Zhang, Zhu Zhu, Chao Liu.

**Writing – review & editing:** Jinmin Cao, Xuan Liu.

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
