## [Decision Letter · Decision Letter 0]

Dear Dr. Jinmin Cao,

Thank you for submitting your manuscript to PLOS ONE. After careful consideration, we feel that it has merit but does not fully meet PLOS ONE’s publication criteria as it currently stands. Therefore, we invite you to submit a revised version of the manuscript that addresses the points raised during the review process.

 Your study addresses a clinically important and understudied area. However, in its current form, several key methodological and reporting issues limit the clarity, interpretability, and scientific rigor of your findings. Reviewer 2 has also identified critical gaps that we ask the authors address in their revision and provide a detailed response. The authors are additionally encouraged to respond to the following queries form the editor

1. Selection of Exposure Measurement: RDW

The manuscript does not indicate when RDW was measured relative to cancer diagnosis and subsequent mortality outcomes. Was RDW assessed at baseline only, or were repeated measurements used? Please clarify the timing and frequency of RDW assessment. Additionally, please provide justification for using RDW as a stable prognostic biomarker in this context, and whether its variability over time was considered.

2. Methodological consideration: Model Building and Covariate Selection

Your three multivariable Cox regression models progressively adjust for a wide range of variables, but the rationale for including specific covariates is not explained. In particular, variables such as HbA1c, HDL-C, and serum creatinine may function as mediators of the RDW–mortality relationship rather than confounders. Please explain the conceptual framework (e.g., directed acyclic graphs or clinical reasoning) guiding your variable selection, and clarify whether model assumptions (such as the proportional hazards assumption) were assessed and met.

3. Handling of Missing Data

The decision to exclude individuals with missing covariate values may introduce bias and reduce generalizability. Please report the extent of missing data for key variables and justify your approach. If feasible, consider applying multiple imputation or conducting a sensitivity analysis to assess whether your findings are robust to missing data assumptions. Also the use of several statistical methods (Cox regression, Kaplan-Meier, RCS, time-dependent ROC) is appreciated but increases the risk of type I error. Please clarify whether any adjustments for multiple comparisons were applied. In addition, we encourage inclusion of model diagnostics, internal validation methods, or sensitivity analyses to evaluate the robustness of your findings.

We look forward to receiving your revised manuscript.

Kind regards,

Danish Ahmad, MBBS,MSc,MNAMS,PhD,IP-FPH(UK),FRCP(Edin),FRCP(Lon)

Academic Editor

PLOS ONE

Journal Requirements:

Reviewers' comments:

Reviewer's Responses to Questions

**Comments to the Author**

1. Is the manuscript technically sound, and do the data support the conclusions?

Reviewer #1: Yes

Reviewer #2: Partly

2. Has the statistical analysis been performed appropriately and rigorously?

Reviewer #1: I Don't Know

Reviewer #2: Yes

3. Have the authors made all data underlying the findings in their manuscript fully available?

Reviewer #1: Yes

Reviewer #2: Yes

4. Is the manuscript presented in an intelligible fashion and written in standard English?

Reviewer #1: Yes

Reviewer #2: Yes

Reviewer #1: On line 114 you say 'available at' but the sentence is incomplete.

The results are interesting, and should be useful to those working in the field.

The English language is clear. The Figures are clear,

Reviewer #2: General

1. This study aims to determine the association between RDW and all-cause or cardiovascular mortality in 476 breast cancer patients in a cohort study using the NHANES data compared to 15,806 patients without breast cancer.

2. RWD alone will hardly be used or useful in the setting of mortality. However, you may want to prominently discuss the strength of your approach: In these times of big daa, RWD may be an inexpensive, simple and seemingly unspecific, yet surprisingly inofrmative parameter during the comprehensive evaluation of paient data to predict outcome.

3. It is refreshing to see how researchers from China help to utilize and analyze data from a US-based survey of 20 years, hinting to the power of publically available datasets. More such crossborder transperancy could benefit all.

4. The English is good and the use of AI to 'polish' the manuscript appreciated (line 373). Please be specific what exact tool(s) have been used (line 374).

5. I would recommend to focus on the immediate observation and shorten the manuscript substantially, particularly its Discussion. This sharpening of the message could enhance the quality for the readers.

Specific

6. I would not call this a "relationship" (line 23). Rather, it is a correlation of a surrogate marker (RDW) with highly clinically relevant outcome parameters. The "strong predictive power" (line 47) is, unfortunately, of little practical utility for now. What would you recommend to do once the RDW is determined? Also: how strong is "strong" and when, compared to which other paraameters, is RWD a stronger parameter?

7. RDW groups (line 189) are at the core of the study and need to be better defined. There may be dynamic changes of RDW at multiple time points. Define when exactly RDW was measured. What RDW did you use if there are several RDW tests with differing results?

8. You document carefully the many parameters (lines 201 to 203, Table 1 and Table S1) that are significantly correlated with RWD. These other parameters, such as age, race, education and so on, may well influence the outome. What exactly does RWD add? Where do you prove the added benefit of testing RWD and how are you taking RWD in account?

9. Line 282: "a likely independent forecaster" - what is the proof of being independence (and independent of what exact other parameters)?

10. RWD is a surrogate marker and iany possible mechanistic link to outcome is lacking. It does not help when you explain the lack of a link by RWD's association with other surrogate markers (lines 292 - 296). Shorten these pseculations much.

11. Your "inability to establish a definitive causal link" (line 352) is not "due to its retrospective cohort design" (line 354). There may well be no definitive causal link at all. You do not make a good case that "prospective studies are necessary" (line 363 - 364) or helpful in any way.

12. RDW is a measure of red cell volume heterogeneity, as such, red cell transfusion can be a confounding factor. How are you taking account of this possible effect? Compare PMID 29770452 and discuss.

Minor

There are many spots were refining and shortening of the text can contribute to focussing the manuscript on the main message.

13. It's not an "RWD level" (line 69), better: RWD parameter or, simply, "RWD".

14. Add new paragraphs at lines 167 and 181.

15. The presentation of the exhibits can be improved: what is the unit of Time (Fig. 2). Particapants (typo), twice (Fig S1). HR (may be obvious to many readers) still needs definition in the footnote (Table 2).

**Do you want your identity to be public for this peer review?** For information about this choice, including consent withdrawal, please see our Privacy Policy

Reviewer #1: No

Reviewer #2: No

---

## [Author Response · Author response to Decision Letter 1]

1 Jun 2025

Editor

1、Response to Editor

Dear Editor,

Thank you very much for your careful review of our manuscript and for raising these important queries. We truly appreciate the opportunity to address your concerns and improve the quality of our work.

We will carefully and thoroughly address each of your queries in a detailed response, providing the necessary clarifications and justifications to ensure the robustness and validity of our study.

Once again, thank you for your time and valuable feedback.

Comment 1: Selection of Exposure Measurement: The manuscript does not indicate when RDW was measured relative to cancer diagnosis and subsequent mortality outcomes. Was RDW assessed at baseline only, or were repeated measurements used? Please clarify the timing and frequency of RDW assessment. Additionally, please provide justification for using RDW as a stable prognostic biomarker in this context, and whether its variability over time was considered.

Response 1.1

We sincerely appreciate the reviewer's insightful comments and constructive questions, which have helped us clarify key methodological aspects of our study. We have carefully addressed each point below to enhance the transparency and rigor of our manuscript.

In the National Health and Nutrition Examination Survey (NHANES), hematologic parameters (including RDW and MCV) were obtained concurrently with questionnaire data (such as the Medical Conditions Questionnaire [MCQ]) during the same examination visit. NHANES is a cross-sectional survey and generally does not reassess participants over time; thus, only baseline RDW values were analyzed. Mortality outcomes were determined via linkage to the National Death Index (NDI) up to December 2019, allowing for survival analysis despite the lack of follow-up laboratory data.

The original text at Line 121-124:

Ascertainment of RDW

Red cell distribution width (RDW) determination was performed based on the full blood count data, utilizing a Beckman Coulter MAXM automated hematology analyzer to measure the values.

Revised Manuscript at Lines 121-125:

Ascertainment of RDW

In our study, RDW was measured only at baseline during the initial examination visit, and no repeated measurements were conducted over time. Red cell distribution width (RDW) determination was performed based on the full blood count data, utilizing a Beckman Coulter MAXM automated hematology analyzer to measure the values.

Comment 1.2: Additionally, please provide justification for using RDW as a stable prognostic biomarker in this context, and whether its variability over time was considered.

Response 1.2

We sincerely appreciate the reviewer's insightful question regarding the use of red cell distribution width (RDW) as a stable prognostic biomarker in our study. Our justification is supported by the following points:

A substantial body of research has highlighted the association between red blood cell distribution width (RDW) and mortality outcomes across diverse patient cohorts, such as surgical patients[1], individuals with type 2 diabetes[2], those suffering from foot ulcers[3], burn victims[4], and critically ill patients[5]. These associations highlight the prognostic potential of RDW and its clinical relevance.

Despite the potential for RDW to fluctuate over time or in response to various physiological conditions (such as: nutritional status, inflammation, or cancer therapy), its prognostic power in chronic conditions like diabetes and cardiovascular disease has been robustly validated. In our referenced literature, studies employing the NHANES database have investigated the association between red blood cell distribution width to albumin level and all-cause mortality among patients with diabetic retinopathy and within the general population[6,7]. The single-time-point measurement of RDW still predicts long-term outcomes, indicating that its risk stratification capability remains stable over time. This is consistent with the design and objectives of our study.

While RDW is a convenient single-time-point biomarker, we recognize that its prognostic accuracy may be affected by unmeasured temporal variations. This limitation is addressed in the discussion section. While RDW is a practical single-time-point biomarker, this study acknowledges that its prognostic utility could be influenced by unmeasured temporal variability. This limitation is addressed in the discussion section (Line 356-359:). Future research could consider incorporating repeated RDW measurements to investigate its time-dependent effects, as the reviewer has appropriately suggested.

(Line 356-359:

Limitation---Thirdly, measuring RDW at a single time point may not capture temporal fluctuations that could influence mortality risk. Furthermore, the study population was derived from NHANES data, which may restrict the generalizability of the findings to other populations with disparate characteristics.)

Comment 2: Methodological consideration: Model Building and Covariate Selection

Your three multivariable Cox regression models progressively adjust for a wide range of variables, but the rationale for including specific covariates is not explained. In particular, variables such as HbA1c, HDL-C, and serum creatinine may function as mediators of the RDW–mortality relationship rather than confounders. Please explain the conceptual framework (e.g., directed acyclic graphs or clinical reasoning) guiding your variable selection, and clarify whether model assumptions (such as the proportional hazards assumption) were assessed and met.

Comment 2.1: Your three multivariable Cox regression models progressively adjust for a wide range of variables, but the rationale for including specific covariates is not explained.

Response 2.1

Thank you very much for your valuable suggestions, which have taught me how to use Directed Acyclic Graphs (DAGs) (Figure 1) to determine the covariates to include in our multivariable model. This approach has helped streamline the number of covariates and improve the overall robustness of our model.

The inclusion of covariates was determined using a multifaceted approach, considering (1) documented results from the literature[8–11], (2) univariate Cox regression analysis results (Table 1) with p < 0.05, and (3) the impact of covariates on the exposure-outcome association, with a threshold set at more than 10%(Table 2). The variables included are detailed below: Age, Sex, Race, Martial status, PIR, BMI, Hypertensive, Cardiovascular disease, angina, heart failure, diabetes, family history of diabetes, Hyperlipidemia, HbA1c, Total cholesterol, High density lipoprotein, Albumin, Hemoglobin, MCV, Serum creatinine, eGFR.

Comment 2.2: In particular, variables such as HbA1c, HDL-C, and serum creatinine may function as mediators of the RDW–mortality relationship rather than confounders.

Response 2.2

Thank you for your insightful comment. In response to your query, we have conducted separate mediation analyses to examine the potential mediating roles of HbA1c, HDL-C, and serum creatinine in the relationship between RDW (red cell distribution width) and all-cause mortality. The results of these analyses are presented in Table 3, Table 4, and Table 5, respectively.

Our findings indicate that these variables do not serve as mediators in the RDW-mortality relationship. The mediation effects were found to be non-significant, suggesting that the associations observed are not mediated through these biochemical markers. They may be more likely to act as confounders or to have indirect associations with the outcome.

Comment 2.3�clarify whether model assumptions (such as the proportional hazards assumption) were assessed and met.

Response 2.3

Thank you for your valuable feedback. In response to your question regarding the assessment of model assumptions, including the proportional hazards assumption, we have conducted a thorough analysis.

We have performed tests to evaluate the proportional hazards assumption for our models, focusing on two types of mortality outcomes: all-cause mortality and cardiovascular disease (CVD) mortality. The results of these tests are summarized in the two tables provided.

For both RDW and GLOBAL variables across both mortality types, the p-values obtained from the proportional hazards tests are well above the conventional threshold of 0.05. Specifically, for all-cause mortality, the p-values are 0.795 for both RDW and GLOBAL (Table 6). For CVD mortality, the p-values are 0.934 for both RDW and GLOBAL (Table 7). These results indicate that we fail to reject the null hypothesis that the proportional hazards assumption holds.

Therefore, based on these findings, we conclude that the proportional hazards assumption is met for the models analyzed. This supports the validity of our model estimates and the interpretations drawn from them.

We appreciate your attention to this critical aspect of our analysis and hope that the additional information provided satisfies your query.

Comment 3. Handling of Missing Data The decision to exclude individuals with missing covariate values may introduce bias and reduce generalizability. Please report the extent of missing data for key variables and justify your approach. If feasible, consider applying multiple imputation or conducting a sensitivity analysis to assess whether your findings are robust to missing data assumptions. Also the use of several statistical methods (Cox regression, Kaplan-Meier, RCS, time-dependent ROC) is appreciated but increases the risk of type I error. Please clarify whether any adjustments for multiple comparisons were applied. In addition, we encourage inclusion of model diagnostics, internal validation methods, or sensitivity analyses to evaluate the robustness of your findings.  

Comment 3.1: The decision to exclude individuals with missing covariate values may introduce bias and reduce generalizability. Please report the extent of missing data for key variables and justify your approach.

Response 3.1

We sincerely appreciate the reviewer's insightful comment regarding missing data of covariates.

In response to the concern regarding the exclusion of individuals with missing covariate values, we have carefully assessed the extent of missing data in our dataset. After excluding cases with missing values, the sample size was reduced to 15,806. Our analysis revealed that the poverty-income ratio (PIR) variable had the highest missing rate (9.042%), while all other covariates demonstrated minimal missingness (<5%) (Table 8). Given that the overall missingness is relatively low (<10%for the most affected variable), we initially employed a complete-case analysis by excluding individuals with missing covariate values. This approach is commonly used in the literature when missing data are minimal, as it helps to maintain the simplicity and interpretability of the analysis. Numerous relevant studies on RDW have employed the complete-case analysis method, directly excluding cases with missing covariate values for statistical analysis[1,6].

To ensure robust results and minimize potential bias, we will implement multiple imputation techniques for missing data handling, followed by comprehensive sensitivity analyses to validate the stability of our findings. In contrast, following the application of multiple imputation, the sample size increased to 22,726. (Table 9)

Comment 3.2: Also the use of several statistical methods (Cox regression, Kaplan-Meier, RCS, time-dependent ROC) is appreciated but increases the risk of type I error. Please clarify whether any adjustments for multiple comparisons were applied. In addition, we encourage inclusion of model diagnostics, internal validation methods, or sensitivity analyses to evaluate the robustness of your findings.

Response 3.2:

In this study, multiple statistical methods, including Cox regression, Kaplan–Meier curves, restricted cubic splines (RCS), and time-dependent receiver operating characteristic (ROC) analyses, were employed to investigate the relationship between red blood cell distribution width (RDW) and all-cause as well as cardiovascular mortality in patients with breast cancer. In the original analyses, cases with missing values were directly excluded to ensure data completeness for the initial assessment (see original Table 1, Table 2, Figure 1, Figure 2, and Figure 4). To assess the robustness of the findings in the presence of missing data, multiple imputation was employed to fill in the missing values. The analyses were then repeated using the imputed datasets to evaluate the consistency and stability of the results (see Table 9, Table 10, Figure 2, Figure 3, and Figure 4).

Baseline characteristics: The analysis revealed that the proportions of breast cancer cases (3.0% in both scenarios), all-cause mortality (13.2% vs. 13.1%), and cardiovascular mortality (4.2% in both) were remarkably consistent between the datasets with missing values excluded and those with multiple imputation applied. In contrast, the mean follow-up time showed a notable difference, being 126.9 months in the excluded dataset versus 115.5 months in the imputed dataset (Table 1 original data VS Table 9). The distributions of breast cancer incidence, all-cause mortality, and cardiovascular mortality were largely consistent between the analyses with missing values excluded and those with multiple imputation applied, underscoring the robustness of the findings.

Cox regression: In the Cox regression analysis of all-cause mortality without multiple imputation (Table 2 original data), the hazard ratio (HR) for RDW in the crude model was 1.39 (95% CI: 1.25–1.54), and after full adjustment in Model 3, it increased to 1.56 (95% CI: 1.35–1.80). For the RDW group, the crude model showed an HR of 2.39 (95% CI: 1.66–3.45) for the high RDW group (≥12.68%) compared to the low RDW group (<12.68%), which attenuated to 2.13 (95% CI: 1.42–3.20) in Model 3.After multiple imputation (Table 10), the HR for RDW in the crude model was lower at 1.24 (95% CI: 1.16–1.34) and slightly increased to 1.25 (95% CI: 1.12–1.38) in Model 3. For the RDW group, the crude model HR for the high RDW group was 1.82 (95% CI: 1.36–2.45), attenuating to 1.46 (95% CI: 1.06–2.02) in Model 3. The trend for cardiovascular mortality after multiple imputation is consistent with that before multiple imputation. Although the specific values have changed, the risk of mortality in the high RDW group remains significantly higher than that in the low RDW group.

RCS: The restricted cubic spline (RCS) analysis visualizing the relationship between RDW and all-cause mortality (Panel A) as well as cardiovascular mortality (Panel B) among breast cancer patients demonstrated a linear association in both the original dataset with missing values excluded and the dataset after multiple imputation (p for non-linearity > 0.05), indicating a consistent trend. (Figure 1 original data VS Figure 2)

Kaplan-Meier: we employed Kaplan–Meier survival curves to elucidate the relationship between RDW and both all-cause and cardiovascular mortality in our breast cancer cohort. Notably, the Kaplan–Meier curves derived from the dataset with missing values excluded (Figure 2 original data) and those from the dataset following multiple imputation (Figure 3) consistently demonstrated that patients in the low RDW group (<12.68%) exhibited significantly higher survival rates compared to those in the high RDW group (≥12.68%). This consistent trend across both analytical approaches underscores RDW's robustness as a prognostic indicator of mortality risk.

Subgroup Analyses: In the subgroup analyses for all-cause mortality stratified by age, BMI, diabetes, and eGFR, no significant interaction effects were observed (p > 0.05) following the handling of missing covariate values and multiple imputation. This indicates consistent results across all stratified analyses. (Figure 3 original data VS Figure 4).

time-dependent ROC: The stability in AUC values for all-cause mortality both before and after multiple imputation suggests that RDW is a robust predictor for this outcome. The predictive ability of RDW for all-cause mortality remains consistent over time, regardless of the data handling method. This consistency underscores the reliability

---

## [Editor Report · Decision Letter 1]

Higher Red Cell Distribution Width (RDW) is Associated with Increased All-Cause and Cardiovascular Mortality in Patients with Breast Cancer: A Retrospective Analysis of NHANES Data (1999-2018)

PONE-D-24-49688R1

Dear Dr. Jinmin

We’re pleased to inform you that your manuscript has been judged scientifically suitable for publication and will be formally accepted for publication once it meets all outstanding technical requirements.

Kind regards,

Danish Ahmad, MBBS,MSc,MNAMS,PhD,IP-FPH(UK),FRCP(Edin),FRCP(Lon)

Academic Editor

PLOS ONE

Additional Editor Comments (optional):

Dear Authors

The revised paper addressed reviewer and my comments satisfactorily allowing the paper to progress to potential publication. The word count, number of tables and figures should match the journal requirements. I am slightly concerned about the authors self-declaration for use of AI to polish the manuscript. While the authors declare in the response to reviewers, I would also expect that a statement of AI use should be produced and reflected in the final paper.

Best

Danish
---

## [Editor Report · Acceptance letter]

PONE-D-24-49688R1

PLOS ONE

Dear Dr. Cao,

I'm pleased to inform you that your manuscript has been deemed suitable for publication in PLOS ONE. Congratulations! Your manuscript is now being handed over to our production team.

Kind regards,

on behalf of

Dr. Danish Ahmad

Academic Editor

PLOS ONE